# Regulation of Wnt Signaling by FOX Transcription Factors in Cancer

**DOI:** 10.3390/cancers13143446

**Published:** 2021-07-09

**Authors:** Stefan Koch

**Affiliations:** 1Wallenberg Centre for Molecular Medicine (WCMM), Linköping University, 58185 Linköping, Sweden; stefan.koch@liu.se; Tel.: +46-132-829-69; 2Department of Biomedical and Clinical Sciences (BKV), Linköping University, 58185 Linköping, Sweden

**Keywords:** Wnt, forkhead, FOX, beta-catenin, transcription factors

## Abstract

**Simple Summary:**

Cancer is caused by a breakdown of cell-to-cell communication, which results in the unrestricted expansion of cells within a tissue. In many cases, tumor growth is maintained by the continuous activation of cell signaling programs that normally drive embryonic development and wound repair. In this review article, I discuss how one of the largest human protein families, namely FOX proteins, controls the activity of the Wnt pathway, a major regulatory signaling cascade in developing organisms and adult stem cells. Evidence suggests that there is considerable crosstalk between FOX proteins and the Wnt pathway, which contributes to cancer initiation and progression. A better understanding of FOX biology may therefore lead to the development of new targeted treatments for many types of cancer.

**Abstract:**

Aberrant activation of the oncogenic Wnt signaling pathway is a hallmark of numerous types of cancer. However, in many cases, it is unclear how a chronically high Wnt signaling tone is maintained in the absence of activating pathway mutations. Forkhead box (FOX) family transcription factors are key regulators of embryonic development and tissue homeostasis, and there is mounting evidence that they act in part by fine-tuning the Wnt signaling output in a tissue-specific and context-dependent manner. Here, I review the diverse ways in which FOX transcription factors interact with the Wnt pathway, and how the ectopic reactivation of FOX proteins may affect Wnt signaling activity in various types of cancer. Many FOX transcription factors are partially functionally redundant and exhibit a highly restricted expression pattern, especially in adults. Thus, precision targeting of individual FOX proteins may lead to safe treatment options for Wnt-dependent cancers.

## 1. Wnt Signaling in Cancer

Wnt signaling is one of the fundamental developmental signaling pathways in the animal kingdom. In adult organisms, it functions as a key regulator of stem cell maintenance and proliferation. Consequently, Wnt signaling is indispensable for numerous biological processes ranging from embryogenesis to tissue homeostasis and regeneration.

The basic principles of Wnt signaling are well-understood [1,2] (Figure 1). In the central Wnt pathway, referred to as Wnt/β-catenin or canonical Wnt signaling, pathway activity is determined by the abundance of the transcription co-factor β-catenin. In unstimulated cells, cytosolic β-catenin is constitutively marked for proteasomal degradation by a multi-protein destruction complex. Upon activation of Frizzled (FZD) family core receptors and Low-density lipoprotein receptor-related protein (LRP) 5/6 co-receptors by secreted Wnt ligands, the destruction complex is inhibited, which allows β-catenin accumulation. β-catenin then translocates to the nucleus, where it drives the expression of target genes of Transcription factor/Lymphoid enhancer-binding factor (TCF/LEF) family transcription factors. This gene expression signature, which includes numerous shared as well as tissue-specific gene targets [3], promotes stem cell maintenance, cell cycle progression, and cell proliferation, and is thus essential for organogenesis and tissue homeostasis. Besides Wnt/β-catenin signaling, there are several non-canonical Wnt pathways that branch off at the receptor level and at the level of the destruction complex [4,5]. In Wnt/Planar cell polarity (PCP) as well as Wnt/Ca^2+^ signaling, Wnt ligands engage FZDs and alternative co-receptors to elicit a β-catenin-independent response. The signaling outcome of these pathways is primarily non-transcriptional, and is best known for regulating cell migration and polarity. Finally, Wnt/Stabilization of proteins (STOP) and Wnt/Target of rapamycin (TOR) signaling promote the stability of proteins other than β-catenin via the Wnt-induced inhibition of the destruction complex, and support proteome maintenance and cell growth in a transcription-independent manner [4].

Given the prominent roles of Wnt signaling in embryogenesis and stem cell regulation, it is unsurprising that dysregulated Wnt pathway activity is associated with numerous serious disorders, especially cancer [1,6,7]. Indeed, the term Wnt itself references the identification of the first mammalian pathway component (Wnt1, then known as int-1) as an oncogene activated by viral integration [8]. In the more than 30 years following this discovery, Wnt/β-catenin signaling in particular has taken center stage as a major oncogenic pathway in various types of cancer. A prime example for Wnt-dependent cancer is sporadic colorectal cancer (CRC). The vast majority of tumors in CRC patients have activating mutations in Wnt pathway components, of which most result in loss-of-function of the destruction complex or the aberrant stabilization of β-catenin [9]. Additionally, elevated Wnt ligand levels increase pathway activity in CRC irrespective of underlying mutations [10]. Chronic Wnt pathway activation in colonic epithelial cells leads to the formation of hyper-proliferative adenomas, which are hotspots for mutations in other genes involved in cancer initiation and progression. Evidence from mouse models of CRC suggests that restoration of normal Wnt activity is sufficient to induce complete tumor regression even in fully transformed colon cancer [11], which highlights the importance of sustained Wnt signaling not only in tumor development, but also cancer stem cell maintenance. Wnt pathway mutations are common in other types of cancer as well, although they occur at a lower frequency compared to CRC. For example, one in four hepatocellular carcinomas and uterine corpus endometrial carcinomas exhibit stabilizing β-catenin mutations, while destruction complex mutations are frequently observed in cutaneous melanomas and gastric adenocarcinomas [6]. Wnt pathway mutations are relatively rare in other cancers, including pancreatic and prostate cancer. Nonetheless, Wnt signaling activity is increased in many of these tumors, which is presumably caused by expression changes in pathway components such as Wnt ligands and receptors [12,13].

Apart from tumor initiation, Wnt signaling also contributes to cancer metastasis [7,14]. A hallmark of advanced carcinomas is epithelial-to-mesenchymal transition (EMT), which enables cancer cells to leave the main tumor and spread to different locations in the body. Several key regulators of EMT are Wnt/β-catenin target genes, including the zinc finger proteins SNAI1 and SNAI2 that repress epithelial cell adhesion molecules [15,16]. Wnt signaling additionally stabilizes SNAI1/2 protein through Wnt/STOP [17,18], resulting in rapid EMT following Wnt pathway activation. Wnt/PCP and Wnt/Ca^2+^ signaling promote metastasis as well, by increasing tumor cell migration and invasion [19,20]. In particular, altered expression of WNT5A and WNT11, which predominantly activate non-canonical Wnt pathways, has been linked to metastasis in various cancers [12,20]. Wnt signaling has additional roles in tumor progression and treatment resistance, but these are less well understood.

In light of these findings, it is obvious that the Wnt signaling pathway is a priority target for therapeutic intervention in cancer, and clinical trials are currently underway to assess the efficacy of Wnt modulators in several types of cancer [21,22]. However, owing to the pervasive functions of Wnt signaling in normal tissue homeostasis, it remains a considerable challenge to minimize the on-target toxicity of these drugs especially in the intestinal epithelium and the hematopoietic system [23]. It is therefore a major goal in preclinical Wnt research to identify context-dependent pathway regulators that can be safely targeted in human diseases.

### Regulation of β-catenin-dependent Gene Transcription

Although the basics of Wnt signaling are well-established, it is considerably less well understood how specific Wnt pathway outputs are generated in different tissues and cell types. For example, whether Wnt signaling elicits canonical or non-canonical downstream effects is mostly determined by the interaction of 19 different Wnt ligands with 10 FZD receptors in humans, but few studies have explored the signaling outcomes of individual ligand/receptor combinations [24,25,26,27]. Similarly, it is poorly understood how tissue-specific target genes can be selectively induced by the β-catenin transcriptional complex. Mounting evidence suggests that a wide array of transcription co-factors fine-tunes Wnt target gene expression [28,29,30], but as Söderholm and Cantù pointed out, the identity and function of such transcriptional regulators has received relatively little attention in the field [31].

The current paradigm in β-catenin-dependent gene transcription is that the four TCF/LEF family transcription factors (TCF7, TCF7L1, TCF7L2, and LEF1) constitutively occupy Wnt-responsive DNA elements, and that they are kept in an inactive state by Transducin-like enhancer (TLE) family repressors. Upon Wnt pathway activation, β-catenin binds to TCF/LEF and inactivates TLE, thereby enabling target gene transcription [32]. Selective gene expression is then achieved by the differential action of the four TCF/LEF transcription factors, which have overlapping but non-identical functions and target genes [33,34,35]. However, it is unlikely that this mechanism alone explains the different Wnt transcriptional signatures that have been observed even in developmentally related tissues [3].

Besides TLE, numerous other proteins engage β-catenin and TCF/LEF to stratify the Wnt transcriptional response [28,31]. These interactors include numerous transcription factors that modulate β-catenin activity and Wnt target gene expression. In the prostate, hormonal activation of the androgen receptor (AR) induces its nuclear translocation, where it competes with TCF/LEF for β-catenin binding. This results in the concomitant inhibition of Wnt signaling and activation of AR signaling, which has important implications for prostate development and cancer [36,37]. Similarly, the pluripotency factor POU domain class 5 transcription factor 1 (POU5F1) sequesters β-catenin in the nucleus of stem cells, which simultaneously inhibits Wnt signaling and induces POU5F1 target gene expression, thereby preventing cell differentiation [38,39]. Conversely, the inflammatory effector protein AP-1 associates with the β-catenin transcriptional complex to activate both AP-1 and Wnt target gene expression, which drives the progression of CRC [40,41,42]. The forelimb-specific T-box transcription factor 3 (TBX3) also activates Wnt signaling by associating with the β-catenin transcriptional complex, and its aberrant expression may contribute to CRC metastasis [43]. Moreover, transcription factors can regulate Wnt signaling activity by other means. In addition to its ability to bind to and compete for β-catenin [44], the SRY-Box transcription factor 9 (SOX9) destabilizes β-catenin via the nuclear translocation of the destruction complex and induction of Mastermind-like protein 2 (MAML2) [45,46], and it induces the expression of Wnt pathway components [47].

## 2. Forkhead Box Family Transcription Factors

The aforementioned examples provide a brief glimpse at the complexity of Wnt signaling at the transcriptional level, and illustrate how tissue and cell-specific Wnt activity is controlled by the action of non-TCF/LEF transcription factors. Below, I review how Wnt signaling is shaped by one of the largest mammalian transcription factor families, the forkhead box (FOX) transcription factors.

FOX proteins are highly conserved transcription factors with 44 family members in humans, not counting five duplicated FOXD4-like genes and the FOXO3 variant FOXO3B [48,49]. They are further subdivided into 19 classes (A through S) based on sequence similarity of their eponymous, DNA-binding forkhead domain. FOX transcription factors are involved in virtually all facets of mammalian biology, including embryogenesis and tissue homeostasis. Accordingly, dysfunction of FOX proteins is associated with severe developmental disorders, autoimmune diseases, and cancer [49,50,51,52]. The forkhead domain has undergone relatively little change during evolution from early metazoan ancestors, and most FOX family members bind to similar DNA motifs with differing affinity [53,54,55]. In contrast, the protein sequences flanking the DNA-binding domain have diverged considerably, and it is generally thought that the specific functions of individual FOX transcription factors are conferred by their interaction with other proteins via these unique regions [50,52]. Consistently, proteomics analyses of 33 FOX family members uncovered distinct sets of interactors for each molecule [56], and functional studies from our group revealed that even closely related FOX proteins can have remarkably different transcriptional activity [57]. Although some FOX transcription factors are expressed near-ubiquitously, most family members exhibit a strongly spatially or temporally restricted expression pattern [49]. For example, FOXE1 is expressed almost exclusively in the thyroid gland, and accordingly, its dysfunction is associated with severe thyroid abnormalities and thyroid cancer [58,59]. Other FOX transcription factors such as FOXB1 and FOXI3 are critical for embryonic development [50], but they are essentially not expressed in adults. Additionally, unlike TCF/LEF transcription factors, FOX proteins are not restricted to any specific signaling pathway, but rather function as downstream effectors in various signaling networks [49].

### 2.1. FOX Transcription Factors Are Wnt Pathway Regulators

Given these considerations, it is not surprising that FOX transcription factors also act as regulators of Wnt signaling. At the time of writing, approximately half of all mammalian FOX transcription factors have been assigned a role in the Wnt pathway, primarily in cancer cells (Table 1). Moreover, in a screening experiment including additional FOX proteins, our group observed that most of them acted as Wnt activators or inhibitors in normal and cancer cells [60]. It appears that the role of FOX proteins in cancer is consistent with their function in the Wnt pathway; that is, FOX proteins that work as Wnt pathway activators generally promote tumorigenesis and are induced in cancers, whereas FOX family members that inhibit Wnt signaling are frequently recognized as tumor suppressors and exhibit decreased expression (Table 1, Figure 2). To what extent the contribution of FOX transcription factors to cancer initiation and progression can be explained by their activity in the Wnt pathway is unclear, and will have to be determined on a case-by-case basis. Nonetheless, it is worthwhile to review how FOX transcription factors can modulate Wnt signaling, considering the importance of dysregulated Wnt signaling in cancer biology.

#### 2.1.1. Regulation of β-catenin Localization and Stability

β-catenin stabilization and nuclear translocation are central events in canonical Wnt signaling [1]. Thus, any molecule that affects the protein level or subcellular localization of β-catenin directly controls the magnitude of the Wnt transcriptional response. Several FOX family members are known to regulate β-catenin in this manner, most prominently the major oncogene FOXM1. FOXM1 is overexpressed in many types of cancer and contributes to all cancer hallmarks, including increased proliferation, reduced apoptosis, tumor metastasis, and drug resistance [88,89,90]. FOXM1 exerts its oncogenic functions through transcriptional as well as non-transcriptional modes of action, which include its interaction with protein partners such as β-catenin [88,91]. Zhang et al. showed that FOXM1 binds β-catenin in normal and cancer cells, and that the Wnt transcriptional activity in different cells correlates with FOXM1 abundance [73]. FOXM1 strongly enhances the nuclear translocation of β-catenin, which is required for the tumor-initiating capacity of glioblastoma cells in vivo (Figure 3a). Subsequent studies confirmed these observations in other cancers such as meningiomas and pancreatic tumors [92,93], suggesting that the nuclear shuttling of β-catenin by FOXM1 is a common pathomechanism in cancer.

Other FOX family members have been implicated in β-catenin stabilization and nuclear import as well. Overexpression of FOXG1 is associated with apoptosis resistance and metastasis in various cancers [94,95]. FOXG1 directly interacts with β-catenin, and depletion of FOXG1 reduced nuclear β-catenin abundance in hepatocellular carcinoma (HCC) cell lines [67]. Consistently, expression of FOXG1 is strongly positively correlated with nuclear β-catenin levels in HCC patient samples. Similarly, Liu et al. reported that FOXJ1 increases nuclear β-catenin levels in CRC cell lines, which the authors attributed to a reduction of truncated (i.e., tumor-initiating) APC in these cells [70]. It is unclear how FOXJ1 could control the abundance of mutant APC, and it should be noted that FOXJ1 is more commonly associated with cancers in which APC mutations are infrequent [96,97]. Finally, FOXQ1 depletion inhibits β-catenin nuclear translocation without affecting β-catenin protein levels in CRC cell lines [98]. FOXQ1 may promote β-catenin nuclear import indirectly via induction of annexin A2 [77], but FOXQ1 also physically interacts with β-catenin in cancer cells [76]. Overexpression of FOXQ1 has been linked to EMT and tumor metastasis in various cancers [99], and thus it is of interest to determine its molecular functions. However, the precise mode by which FOXQ1 and FOXJ1 activate Wnt signaling, and whether this function is relevant for their role in cancer biology, requires further investigation.

Conversely, other FOX transcription factors may inhibit Wnt signaling by destabilizing β-catenin. Higashimori et al. reported that FOXF2 induces the expression of the E3 ubiquitin ligase IRF2BPL in gastric cancer cell lines [80]. IRF2BPL associates with β-catenin and causes its proteasomal degradation independently of the destruction complex. Consistently, FOXF2 inhibited the growth of gastric tumor cells in vivo, and epigenetic silencing of FOXF2 is associated with poor prognosis in stomach cancer patients. FOXF2 additionally induces Secreted frizzled-related protein 1 (SFRP1) in intestinal fibroblasts [81]. SFRP family proteins function as soluble decoy receptors for Wnt ligands [100], and destabilize β-catenin by increasing the activity of the destruction complex. Thus, it is likely that FOXF2 attenuates Wnt signaling via the transcriptional regulation of multiple β-catenin inhibitors.

#### 2.1.2. Regulation of the Wnt Transcriptional Complex

Non-TCF/LEF transcription factors shape specific Wnt signaling outputs by engaging the Wnt transcriptional complex [31], and several studies have shown that this is the case for FOX transcription factors as well. Aside from its role in β-catenin shuttling, FOXM1 reinforces Wnt target gene expression at the chromatin level. Zhang et al. reported that FOXM1 is required for the efficient recruitment of β-catenin to TCF7L2 at Wnt-responsive DNA elements, and thereby increases Wnt target gene transcription [73]. In a follow-up study, the same group showed that FOXM1 facilitates β-catenin/TCF interaction by protecting β-catenin from Beta-catenin-interacting protein 1 (CTNNBIP1)-mediated inhibition [72]. CTNNBIP1 is a ubiquitously expressed protein that blocks TCF activation by competitive binding to β-catenin [101]. FOXM1 interacts with the same β-catenin domain as CTNNBIP1, and thereby de-represses β-catenin/TCF association [72,73].

A different mode of action has been described for FOXP1, which has contrasting functions in various types of cancer [102]. In lymphomas, where its overexpression is associated with poor prognosis, FOXP1 increases Wnt/β-catenin signaling following upstream pathway activation [74]. Walker et al. observed that FOXP1 interacts with β-catenin, TCF7L2, and the acetyl transferase CREB-binding protein (CBP) at TCF binding sites, and that it facilitates the CBP-dependent acetylation of β-catenin (Figure 3b). β-catenin acetylation is an activating post-translational modification that increases the induction of a subset of Wnt target genes [103]. Consistently, FOXP1 was unable to promote Wnt-dependent transcription when the CBP acetylation site on β-catenin was deleted [74]. In their initial screening experiments, Walker et al. also identified the closely related FOXP4 as a potential Wnt activator, but this observation was not explored further. While it thus remains to be tested if FOXP4 controls Wnt signaling as well, a separate study revealed that FOXP3 increases Wnt activity in non-small cell lung cancer (NSCLC) [75]. Yang et al. showed that FOXP3 interacts with β-catenin and TCF7L2, and that FOXP3 overexpression increased β-catenin/TCF association and Wnt target gene expression, similar to FOXP1. Accordingly, FOXP3 gain-of-function increased tumor growth and metastasis of NSCLC cell lines in vivo [75]. FOXP2 was also recently found to engage β-catenin and regulate the expression of several target genes [104], but the functional relevance of this interaction was not investigated in this study. Collectively, these observations suggest that all class P FOX proteins act as Wnt pathway activators. It remains to be determined if this function involves a shared molecular mechanism, such as the promotion of β-catenin acetylation.

FOXG1 enhances Wnt signaling at the chromatin level as well. As with FOXM1 and FOXP1, FOXG1 binds TCF7L2 in addition to β-catenin, and depletion of FOXG1 decreased the recruitment of β-catenin to TCF binding sites in HCC cell lines [67]. The authors of this study did not investigate how FOXG1 promotes β-catenin/TCF interaction, but it has been shown that FOXG1 binds to a different region of β-catenin than FOXM1 and CTNNBIP1 [67,73]. It therefore appears likely that FOXG1 has a distinct mode of action in Wnt transcriptional activation. Finally, FOXH1 promotes the transcription of β-catenin target genes during early *Xenopus* development [68]. Afouda et al. observed that a substantial number of genomic loci were co-occupied by β-catenin and FOXH1 in zygotes. Accordingly, FOXH1 depletion in blastula stage *Xenopus* embryos reduced the expression of maternal Wnt target genes, but this was not associated with altered recruitment of β-catenin to TCF binding sites. Thus, the cooperative action of β-catenin/TCF and FOXH1 transcriptional complexes may be required for the induction of specific Wnt targets, at least during embryonic development.

On the other hand, some FOX transcription factors compete with TCF/LEF for β-catenin binding, and act as inhibitors of canonical Wnt signaling. FOXO3 and FOXO4 engage β-catenin via the same domains as TCF/LEF, and thereby inhibit Wnt transcriptional activity while at the same time increasing the expression of FOXO target genes (Figure 3c) [85,105,106,107]. Unlike most other FOX proteins, the nuclear localization and activity of class O FOX transcription factors is regulated by post-translational modification upon growth factor signaling and oxidative stress [108]. Accordingly, β-catenin was shown to be required for FOXO-dependent stress response and longevity in nematodes [106]. Although these earlier studies focused on cell stress, the reciprocal regulation of Wnt signaling and FOXO-dependent gene transcription is important for cancer biology as well. For example, FOXO3 engages β-catenin and decreases β-catenin/TCF7L2 interaction in prostate cancer cell lines [86]. Liu et al. reported that depletion of FOXO3 induced cancer cell migration, invasion, and EMT, which was blocked by concomitant β-catenin loss-of-function. In other types of cancer, β-catenin may subvert the role of FOXO3 as a tumor suppressor. Induction of FOXO target genes results in cell cycle arrest and apoptosis, which is inhibited by activating mutations in the Phosphoinositide 3-kinase/Protein kinase B (PI3K/PKB) pathway that sequester FOXO in the cytosol [109]. Several clinically used PI3K/PKB inhibitors promote FOXO nuclear translocation, thereby restoring its tumor suppressor function. However, Tenbaum et al. showed that the treatment outcome of these drugs in CRC depends on the abundance of β-catenin [110]. In tumor cells with low β-catenin levels, FOXO3 nuclear translocation caused by PKB inhibitor treatment induced apoptosis and reduced tumor growth in vivo. In cells with high nuclear β-catenin levels, however, PKB inhibition had no effect on cancer cell proliferation, but rather resulted in tumor metastasis. The authors observed that β-catenin activates the expression of specific FOXO3 target genes that are thought to drive cell junction disassembly and migration [110]. Thus, competitive binding of FOXO3 to β-catenin may switch CRC cells from Wnt-dependent proliferation to FOXO-mediated EMT, ultimately causing tumor cell dissemination. Whether these effects are specific for FOXO3 is unclear at this point, but there is some indication that individual FOXO proteins have distinct β-catenin-dependent target genes. Doumpas et al. reported that activation of β-catenin in cells lacking all four TCF/LEF transcription factors results in the differential expression of dozens of target genes that are presumably regulated by other transcription factors [111]. The authors identified FOXO4 as one of these alternative transcription factors, whereas FOXO3 had no effect on gene regulation in the absence of TCF/LEF. Interestingly, FOXO1 disrupts β-catenin/TCF7L2 interaction and inhibits Wnt signaling in pancreatic adenocarcinoma cell lines, but in contrast to FOXO3/4, this could be an indirect effect [84]. Ling et al. reported that FOXO1 induces the long non-coding RNA LINC01197, which binds and inhibits β-catenin. Consistently, LINC01197 overexpression inhibited tumor growth in vivo, and low expression of this RNA was associated with shorter survival in pancreatic cancer patients.

Lastly, FOXN3, a known tumor suppressor in various types of cancer [112], also inhibits Wnt signaling by competitive binding of β-catenin. Similarly to FOXO3/4, FOXN3 overexpression decreases β-catenin/TCF7L2 interaction in CRC cell lines, and depletion of FOXN3 increased tumor growth and metastasis in mice [83]. Collectively, these observations suggest that engagement of the Wnt transcriptional complex is a common mechanism by which FOX proteins regulate Wnt signaling in a context-dependent manner.

#### 2.1.3. Regulation of Wnt Ligand Expression

Another way in which FOX proteins control Wnt signaling is via the transcriptional regulation of Wnt ligands. Induction of Wnt ligand expression has been linked to disease progression in various cancers, such as ovarian, prostate, and lung cancer [113,114,115]. It has been suggested that Wnt pathway activation occurs via the de-repression of Wnt ligand transcription [116], or the epigenetic silencing of secreted Wnt inhibitors such as Dickkopf-related protein 1 (DKK1) and SFRPs [117]. However, it is likely that aberrant transcription factor activity contributes to elevated Wnt ligand levels in cancer. Our group observed that FOXB2, whose expression is normally restricted to the embryonic brain [118], is re-expressed in aggressive prostate cancer subtypes [60]. We found that FOXB2 does not engage β-catenin, but rather activates Wnt signaling via the induction of multiple Wnt ligands (Figure 3d). Inhibition of WNT7B signaling was sufficient to block the effect of FOXB2, and loss-of-function of FOXB2 reduced Wnt transcriptional activity in prostate cancer cell lines. Accordingly, a FOXB2/WNT7B gene expression signature was associated with poor prognosis in prostate cancer patient data. Although our investigation focused on WNT7B, we observed that FOXB2 significantly increased the expression of 14 out of the 19 human Wnt ligands across multiple cell lines. This suggests that FOX transcription factors might work as universal regulators of Wnt ligand expression. In agreement with this hypothesis, FOXQ1 was found to induce the expression of at least three canonical Wnt ligands in mesenchymal stem cells, namely WNT1, WNT3, and WNT7A [77], which are also controlled by FOXB2 [60]. To what extent these ligands contribute to the activity of FOXQ1 in Wnt/β-catenin signaling and cancer biology is unknown, but it is feasible that they amplify the effect on β-catenin nuclear translocation and thereby increase the oncogenic potential of FOXQ1 [76,77].

Other studies have reported the regulation of individual Wnt ligands by FOX transcription factors. FOXC2, a known oncogene in several types of cancer [119], induces WNT4 in myoblast cell lines [64]. Gozo et al. reported that FOXC2 activates Wnt/β-catenin and Wnt/PCP signaling via WNT4, which was required for the inhibition of myoblast differentiation. Concomitant overexpression of both FOXC2 and WNT4 has been observed in some cancers, including CRC [120,121], suggesting FOXC2-mediated induction of WNT4 in tumor cells as well. The other class C FOX protein, FOXC1, also regulates Wnt ligand expression. Han et al. observed that FOXC1 induces the expression of WNT5A in triple-negative breast cancer (TNBC) cell lines, and that WNT5A mediates the FOXC1-dependent effects on cancer cell migration and invasion, but not cell proliferation [62]. The authors further determined that WNT5A induces Matrilysin (MMP7) through a non-canonical Nuclear factor (NF)-κB signaling pathway, which drives tumor metastasis in mice.

Additional FOX proteins may regulate WNT5A expression. Ma et al. reported that protein levels of FOXE1 and the sonic hedgehog signaling pathway effector GLI2 are concomitantly increased in papillary thyroid cancer (PTC) samples [65]. FOXE1 is a transcriptional target of GLI2 [122], and accordingly, depletion of GLI2 in PTC cell lines decreased FOXE1 levels and the expression of WNT5A. Re-expression of FOXE1 in this context restored WNT5 levels, suggesting that GLI2 controls WNT5A via FOXE1 in PTC [65]. Conversely, increased WNT5A expression has been observed in intestinal mesenchymal cells of FOXF2-deficient as well as FOXF1/2 compound heterozygous mice [79]. Ormestad et al. observed that FOXF1/2, which are also controlled by sonic hedgehog signaling, negatively regulate the expression of WNT5A via induction of Bone morphogenetic protein 4 (BMP4). Accordingly, loss of FOXF expression in embryonic mice resulted in epithelial overgrowth in the small intestine, presumably due to WNT5A-dependent β-catenin signaling activation. Furthermore, FOXG1 represses WNT8B expression and Wnt/β-catenin signaling in *Xenopus* embryogenesis, again working as a downstream effector in the sonic hedgehog signaling pathway [66]. It should be noted that FOXG1-deficient mice exhibit severe developmental defects, but no changes in WNT8B expression [123], possibly indicating mammalian-specific functions of FOXG1. Collectively, these studies suggest that FOX transcription factors act as Wnt ligand regulators during tissue patterning and organogenesis, and that the aberrant reactivation of these developmental programs may contribute to cancer development and progression.

#### 2.1.4. Other Mechanisms of Wnt Pathway Regulation

Besides Wnt ligands, β-catenin itself may be a transcriptional target of some FOX proteins. FOXS1 overexpression decreased the transcription of β-catenin (encoded by the CTNNB1 gene) and two Wnt target genes in gastric cancer cell lines, which was associated with decreased tumor cell proliferation in mice [87]. These findings are consistent with the observation that high expression of FOXS1 is linked to improved relapse-free survival in breast cancer patients, and that FOXS1 is highly expressed in the hedgehog but not Wnt subtype of medulloblastomas [124]. In contrast, FOXH1 increased the mRNA levels of CTNNB1 and Wnt target genes in breast cancer cell lines, and depletion of β-catenin reversed the positive effects of FOXH1 on tumor cell proliferation and invasion [69]. Cao et al. observed that FOXC1 also increases CTNNB1 expression in lung cancer cell lines, and identified a forkhead box binding site in the CTNNB1 promoter that was required for the efficient activation of β-catenin transcription [61]. Additionally, FOXC1 decreases the levels of Wnt inhibitor DKK1 in gastric cancer cell lines by direct transcriptional repression [63]. DKK1 attenuates Wnt signaling by competitive binding to LRP5/6, which blocks pathway activation and thereby reduces β-catenin protein levels [125]. Consistently, decreased tumor cell proliferation caused by FOXC1 depletion was reverted by concomitant knock-down of DKK1 [63]. FOXM1 was also recently shown to induce DKK1 in pancreatic and esophageal carcinoma cells [126]. However, Kimura et al. reported that DKK1 does not regulate Wnt signaling in these cells, but rather increases FOXM1 levels in positive feedback loop. Taken together, these results suggest that FOX transcription factors control β-catenin abundance both transcriptionally and post-translationally. At least in the case of FOXC1, these modes of action appear to work in parallel with potentially additive effects.

FOXK1 and FOXK2 have a unique mode of action in the Wnt pathway. FOXK1/2 are best known as metabolic regulators, but they have also been implicated in the pathogenesis of various types of cancer [127,128,129]. Wang et al. reported that FOXK1/2 directly interact with the Segment polarity protein disheveled homologs (DVL) 1–3, and facilitate their nuclear translocation [71]. DVLs are scaffold proteins required for destruction complex inhibition in the Wnt pathway [130]. In addition, nuclear DVL promotes β-catenin/TCF association, and thereby increases Wnt transcriptional activity [40,131]. Consistently, Wang et al. observed that FOXK1/2 overexpression increased Wnt/β-catenin signaling, which was blocked by DVL depletion [71]. High FOXK1/2 levels are associated with increased Wnt target gene expression in CRC patient samples, and depletion of FOXK in CRC cell lines considerably attenuated tumor growth in vivo. Conversely, overexpression of FOXK2 in mice induced epithelial hyper-proliferation and increased Wnt transcriptional activity in intestinal crypts.

Lastly, FOXL1 may regulate Wnt signaling through a distinct mechanism as well. Perreault et al. observed that epithelial cell proliferation was increased in the digestive tract of FOXL1-deficient mice [82]. This increase was associated with elevated nuclear β-catenin levels, which the authors attributed to the post-transcriptional de-repression of the heparan sulfate proteoglycans (HSPG) syndecan-1 and perlecan following loss of FOXL1. HSPGs are common co-receptors in cell signaling pathways, including Wnt signaling [5,132], and thus FOXL1 may attenuate epithelial cell proliferation by curbing Wnt agonist sequestration by HSPGs. Consistent with this role as a putative tumor suppressor, loss of FOXL1 has been linked to tumor progression and poor prognosis in several types of cancer, including pancreatic and gastric cancer [133,134].

### 2.2. Reciprocal Regulation of FOX Transcription Factors by Wnt Signaling

TCF/LEF binding sites are highly abundant in the mammalian genome. It has been shown that Wnt/β-catenin signaling controls the expression of thousands of genes [111,135], and some FOX transcription factors have been identified as *bona fide* Wnt targets as well. Christensen et al. reported that FOXQ1 is one of the most highly upregulated genes in CRC, and that its expression correlates with Wnt pathway activity in CRC cell lines [136] (Figure 2). The authors demonstrated that β-catenin directly engages and activates the FOXQ1 gene promoter, which was attenuated by mutation of a TCF7L2 binding site. FOXQ1 is a potent activator of Wnt/β-catenin signaling [60,77], and thus FOXQ1 induction may trigger a positive feedback loop that promotes tumor progression in some cancers, including CRC. Similarly, β-catenin controls FOXA2 expression by direct transcriptional regulation, and high FOXA2 levels were observed in endometrial hyperplasia samples with elevated Wnt activity [137]. Class A FOX proteins are pioneer transcription factors that act as essential co-factors for various other transcription regulators, such as sex hormone receptors. Accordingly, FOXA2 has been linked to the pathogenesis of prostate and liver cancer [138,139], which could be explained in part by dysregulated Wnt signaling. Others have demonstrated the regulation of FOXC1/2, FOXD3, FOXJ1, and FOXN1 expression by β-catenin/TCF in different contexts [140,141,142,143], but the relevance for cancer biology is less clear in these cases.

Finally, a separate mode of reciprocal regulation has been described for FOXM1. Chen et al. identified FOXM1 as a target of Wnt/STOP signaling [72]. In Wnt-inactive cells, FOXM1 is marked for proteasomal degradation by the ubiquitin ligase F-box/WD repeat-containing protein 7 (FBXW7) following phosphorylation by Glycogen synthase kinase-3 (GSK3) within the destruction complex. Upon Wnt pathway activation, the destruction complex is inhibited, which results in the concomitant stabilization of β-catenin and FOXM1. FBXW7 is a major tumor suppressor that is frequently mutated in cancer [144], and part of its effect can likely be attributed to the post-translational repression of FOXM1. Whether Wnt/STOP controls the stability of other FOX transcription factors is unknown, but several FOX proteins, including FOXA1/3, FOXI1, and FOXK2 contain GSK3/FBXW7 recognition motifs, and may be subject to Wnt-dependent stability regulation.

### 2.3. Common Themes and Open Questions

As discussed in the preceding chapters, there is mounting evidence that FOX transcription factors are common regulators of Wnt signaling pathways, particularly Wnt/β-catenin signaling. Given that both β-catenin and FOX proteins can be traced back to early metazoan ancestors [145,146], it is conceivable that the reciprocal regulation of FOX and Wnt arose during the evolution of multicellular organisms, and that it is conserved throughout the animal kingdom. Support for this hypothesis comes from the observation that the functions of some FOX transcription factors in the Wnt pathway are shared between mammals and lower organisms, such as the Wnt inhibitory role of FOXO proteins in nematodes [106], or the activation of β-catenin target genes by FOXH1 in frogs [68,69]. These examples also illustrate that both positive and negative regulation of Wnt signaling by FOX proteins have likely appeared early in evolution. What could be the purpose of Wnt pathway regulation by FOX transcription factors? One likely possibility is that is has emerged to control the precise temporal and spatial activity of Wnt signaling throughout an organism. With increasing organismal complexity, ever more precisely controlled Wnt activity is required for tissue development and homeostasis. Unlike the core Wnt pathway components, which have diversified little during evolution [147], the FOX transcription factor family continues to evolve, and its members exhibit highly distinct expression patterns [48,49]. Thus, FOX proteins are ideally suited to act as rheostats of Wnt activity in specific cell types, tissues, and developmental stages. It is therefore an open challenge to determine to what extent and in which contexts FOX transcription factors contribute to the regulation of Wnt signaling.

Another unanswered question is whether findings for one FOX protein can be extrapolated to other family members. As outlined above, multiple FOX transcription factors interact with β-catenin and thereby regulate its stability, activity, and subcellular localization (Figure 4). It is not clear if this is a conserved feature in the FOX family, or whether it arose independently in different family members through convergent evolution. Zhang et al. mapped the minimal β-catenin-binding sequence of FOXM1 to its highly conserved forkhead box [73], which suggests that the DNA-binding domain of FOX transcription factors may be a shared β-catenin interaction surface. However, it has been shown that the forkhead box of FOXO1 is dispensable for β-catenin binding [106]. Moreover, in work from our group, we did not observe interaction of full-length FOXB2 with β-catenin [60]. Conversely, β-catenin engages the structurally unrelated FOXG1 and FOXO1 via its N-terminal, and FOXM1 through its C-terminal Armadillo repeats [67,73,106], again arguing against a conserved mode of action.

On the other hand, the role of FOX proteins in the transcriptional regulation of Wnt pathway components may be underappreciated. Our observation that FOXB2 regulates the expression of most Wnt molecules raises the possibility that the majority of Wnt ligands are under transcriptional control by FOX proteins [60]. Many FOX transcription factors have overlapping DNA recognition motifs, and therefore act partly redundantly in target gene expression [53,54]. Consistently, FOXQ1 was shown to regulate some of the same Wnt ligands as FOXB2 [77], and WNT5A transcription is regulated by FOXC1, FOXE1, and FOXF1/2 in different contexts [62,65,79]. Additionally, motif analysis of the FOXC1 binding site that Cao et al. identified in the CTNNB1 promoter suggests that various other FOX proteins may engage this site with similar affinity, and thereby control Wnt/β-catenin signaling transcriptionally [61]. Thus, it is of considerable interest to further investigate the regulation of Wnt pathway genes by FOX transcription factors.

It should be noted that many of the studies referenced here have limitations that need to be considered when interpreting the results. Firstly, FOX proteins are often studied by ectopic overexpression, which may cause them to work in ways that are not physiologically relevant. Secondly, in most cancers, multiple FOX family members are dysregulated at the same time (see Figure 2), which may have combinatorial effects on the Wnt pathway. Lastly, regulation of Wnt signaling is just one the many functions of FOX transcription factors. Thus, whether an observation such as tumor growth following FOX induction can actually be attributed to Wnt pathway activation is usually hard to distinguish from alternative explanations. Clearly, a lot of work still lies ahead in determining how the regulation of Wnt signaling by FOX proteins contributes to cancer biology. However, the effort seems to be well worth it considering the exciting discoveries that have been made so far.

### 2.4. Therapeutic Targeting of FOX-dependent Wnt Signaling

Could the FOX-dependent regulation of Wnt signaling be targeted for cancer therapy? While it is too early to answer this question with confidence, there are some promising results from preclinical studies that give reason to be optimistic. As discussed earlier, high nuclear β-catenin levels confer apoptosis resistance to colorectal cancer cells treated with PI3K/PKB inhibitors [110]. Simultaneous inhibition of PKB and Wnt signaling reverts this effect, and suppresses tumor growth in patient-derived cancer cells [110,148]. Moreover, a recently developed FOXM1 inhibitor reduces β-catenin abundance and activity in different cancer cell lines, and attenuates tumor growth in mouse xenograft models [149]. In the context of Wnt signaling, there are drugs that block the interaction between TCF and β-catenin or CBP, of which some have already entered clinical trials [22]. It is conceivable that similar compounds could target the interaction of FOX proteins with Wnt pathway components, which may aid in the safe and precise treatment of Wnt-dependent cancers.

## 3. Conclusions

The Wnt pathway and FOX transcription factors are ancient regulators of metazoan cell signaling, and their interaction is intimately linked not only to normal development and tissue homeostasis, but also to cancer initiation and progression. Considering the sheer number of FOX family members, and the fact that many of them have not been studied in great detail, it is highly likely that many important connections in FOX-dependent Wnt signaling still await discovery, and that these findings will further refine our understanding of cell signaling in health and disease.

## Figures and Tables

**Figure 1 cancers-13-03446-f001:**
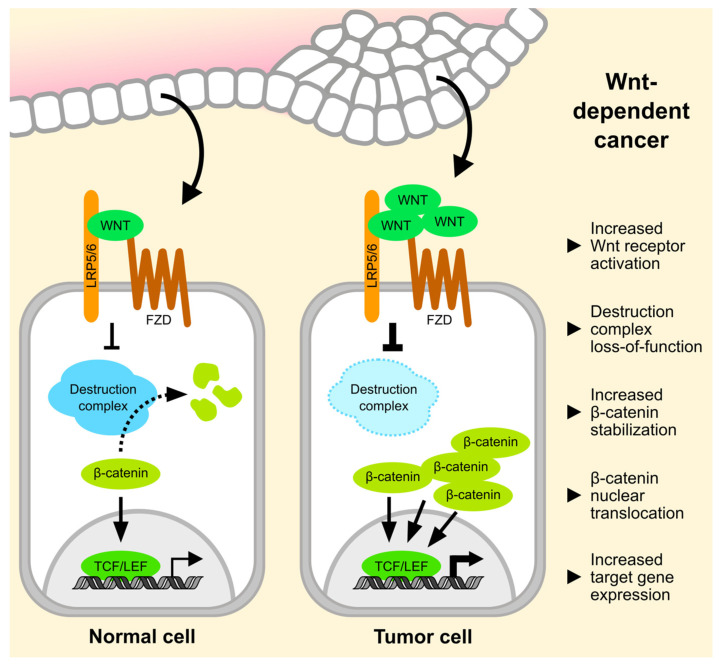
Overview of the canonical Wnt/β-catenin signaling pathway. Hallmark alterations in Wnt pathway activity in Wnt-dependent cancer cells are indicated on the right.

**Figure 2 cancers-13-03446-f002:**
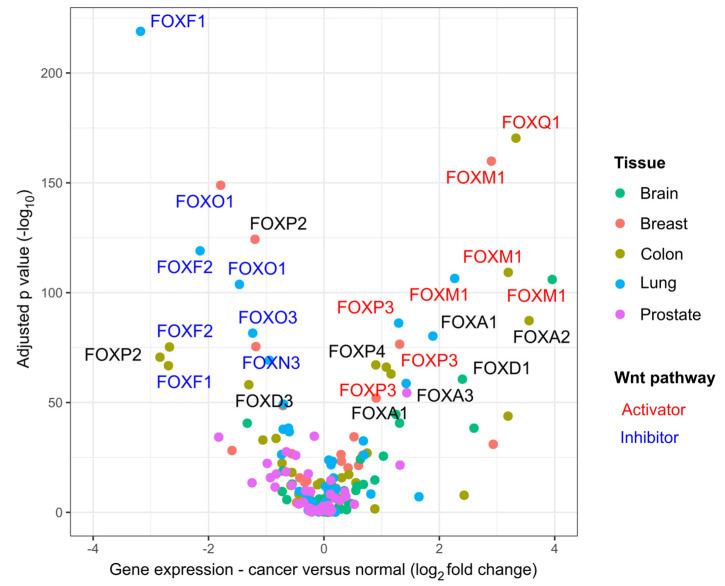
Gene expression changes of FOX transcription factors in selected cancers. Data sources are indicated in the data availability statement at the end of the manuscript. Classification of genes as Wnt pathway activators or inhibitors is based on the literature, see Table 1. Only data points with >2-fold change in expression and an adjusted *p* value < 10^−50^ are labeled.

**Figure 3 cancers-13-03446-f003:**
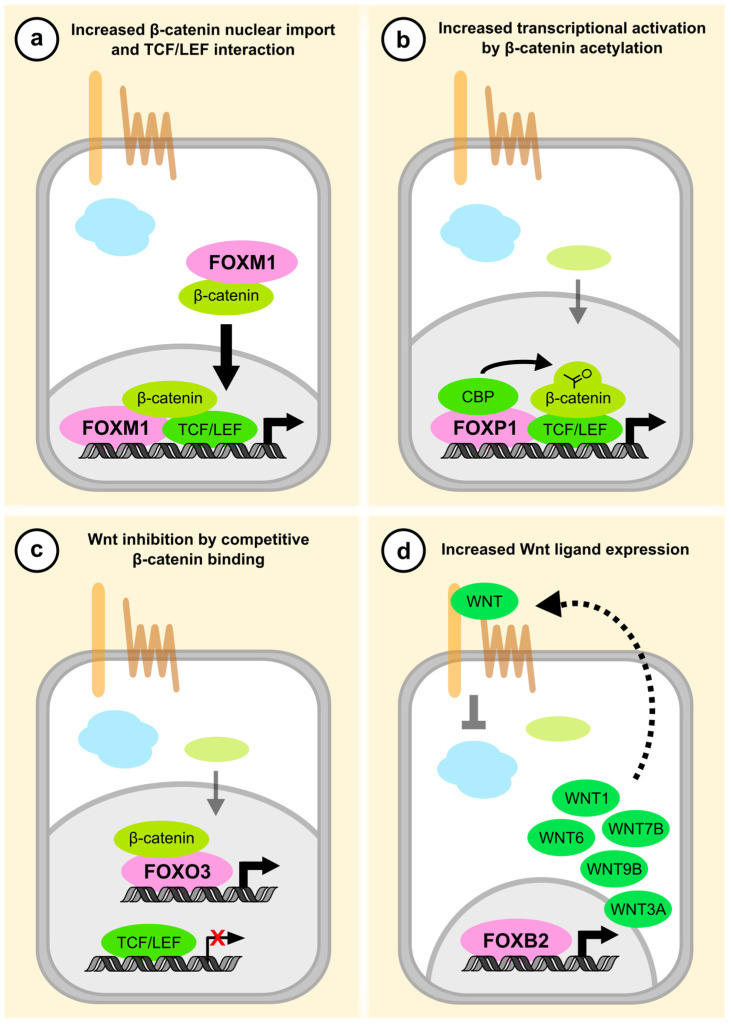
Proposed mode of action of selected FOX transcription factors in the Wnt signaling pathway.

**Figure 4 cancers-13-03446-f004:**
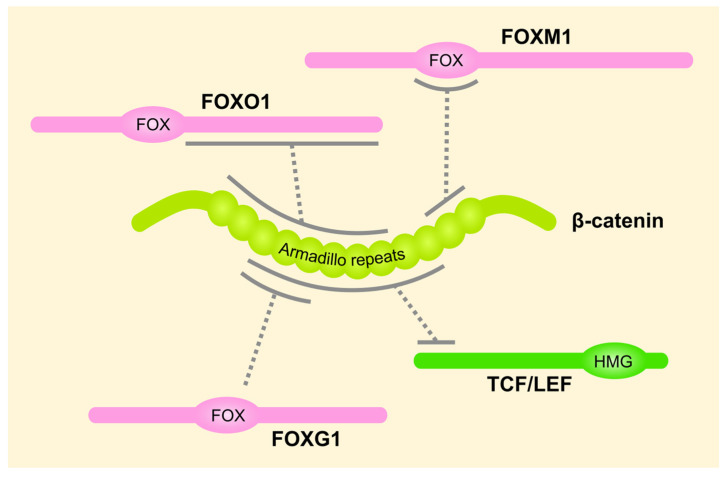
Schematic representation of known FOX protein/β-catenin interactions. Solid grey lines indicate experimentally mapped binding domains. All proteins are shown in N-to-C-terminal orientation. FOX, (DNA-binding) forkhead box; HMG, (DNA-binding) high mobility group box.

**Table 1 cancers-13-03446-t001:** Overview of FOX transcription factors linked to both cancer biology and Wnt signaling.

FOX	Relevance for Cancer Biology	Proposed Function in Wnt Signaling	Refs.
*Wnt pathway activators*			
FOXB2	Putative oncogene in prostate cancer	Induces various Wnt ligands	[60]
FOXC1	Increases proliferation and metastasis in many cancer types	Induces CTNNB1 ^1^ and WNT5A; represses Wnt inhibitor DKK1	[61,62,63]
FOXC2	Increases proliferation and metastasis in many cancer types	Induces WNT4	[64]
FOXE1	Mutation associated with thyroid cancer	May induce WNT5A	[65]
FOXG1	Associated with cancer progression in glioma and hepatocellular carcinoma	Binds and stabilizes β-catenin; promotes TCF7L2 activity; inhibits Wnt ligand expression during development	[66,67]
FOXH1	Drives cell proliferation in acute myeloid leukemia	Synergizes with β-catenin in target gene expression; may induce CTNNB1	[68,69]
FOXJ1	Associated with tumor progression in many cancer types	May stabilize β-catenin via inhibition of APC	[70]
FOXK1	Increases proliferation and metastasis in many cancer types	Promotes nuclear translocation of Wnt scaffold protein DVL	[71]
FOXK2	Context-dependent oncogene or tumor suppressor	Promotes nuclear translocation of Wnt scaffold protein DVL	[71]
FOXM1	Major oncogene in many cancer types	Promotes β-catenin nuclear translocation; stabilizes β-catenin/TCF7L2 interaction; synergizes in target gene expression	[72,73]
FOXP1	Context-dependent oncogene or tumor suppressor	Activates β-catenin via CBP-dependent acetylation; synergizes in target gene expression	[74]
FOXP3	Context-dependent oncogene or tumor suppressor	Synergizes with β-catenin/TCF7L2 in target gene expression	[75]
FOXQ1	Promotes metastasis in carcinomas; tumor suppressor in melanomas	Induces Wnt ligands; may promote β-catenin nuclear translocation via annexin A2	[76,77]
FOXR2	Oncogene in many cancer types	Unclear	[78]
*Wnt pathway inhibitors*			
FOXF1	Context-dependent oncogene or tumor suppressor	May inhibit WNT5A	[79]
FOXF2	Tumor suppressor in gastric and cervical cancer	May inhibit WNT5A and induce Wnt inhibitor SFRP1; promotes β-catenin degradation	[79,80,81]
FOXL1	Tumor suppressor in multiple types of cancer	May reduce proteoglycan co-receptor levels	[82]
FOXN3	Inhibits proliferation and migration in multiple types of cancer	Inhibits β-catenin/TCF7L2 interaction	[83]
FOXO1	Tumor suppressor in multiple types of cancer	Inhibits β-catenin/TCF7L2 interaction, possibly via LINC01197	[84]
FOXO3	Tumor suppressor in multiple types of cancer	Inhibits β-catenin/TCF7L2 interaction by competitive binding	[85,86]
FOXO4	Tumor suppressor in multiple types of cancer	Inhibits β-catenin/TCF7L2 interaction by competitive binding	[85]
FOXS1	Putative tumor suppressor in breast cancer and hepatocellular carcinoma	May inhibit CTNNB1 expression	[87]

^1^ Gene encoding β-catenin.

## Data Availability

Figure 2 was generated using public gene expression data from The Cancer Genome Atlas (TCGA Research Network). Cancer and matched normal data were accessed and analyzed in GEPIA 2 (http://gepia2.cancer-pku.cn/#index, accessed on 10 June 2021) [150], using the ANOVA differential expression option. The following TCGA datasets were used: GBM (Brain), BRCA (Breast), COAD (Colon), LUAD (Lung), and PRAD (Prostate). Results were visualized using R 4.1.0 [151].

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
