# Peer review of "Regulation of Wnt Signaling by FOX Transcription Factors in Cancer"

_cancers, 2021, doi:10.3390/cancers13143446_

Round 1
Reviewer 1 Report
Excellent review that covers a wide array of literature concerning the interaction between various members of the FOX family and the WNT signalling pathway. In general, it is well balanced and as such will provide a great help for all that want to study this further. Being this the case I think (personally) that the reader would also be helped by being more critical concerning the literature. Now there is a "common themes and open questions" section which is laudable but could be strengtheedn for example by "cautionary notes" . For example, it is now suggested that because FOX family members bind to a similar DNA binding consensus many FOXO members could regulate expression of WNT modulators and therefore " the role of FOX proteins in the transcriptional regulation of Wnt pathway components may be underappreciated". However and alternatively one could argue that it is overestimated because many approach gene regulation by overexpression of FOX components and this may force binding to these same DNA elements. Also because e.g. FOXM1 regulates proliferation it should be considered that proliferation per se regulates many of the observed changes rather than these being FOXOM1 specific, same holds for FOXOs that can impose cell cycle arrest. Although have not gone to such detail through many of the referred papers I will not be surprised if several studies do not control for this possibility.
In addition, many studies lack details on the interaction e.g. between FOX and beta-catenin and for example the interaction between FOXM1 and beta-catenin has been suggested to at least involve the conserved FH domain. This may however also imply that if FOX proteins employ this mode of binding beta-catenin in general, this binding may inhibit FOX binding to DNA and hence transcriptional activity. So, could beta-catenin be considered a general FOXO repressor?
So if anything I would encourage the author to include a little more critical appraisal of literature to help new-comers to the field on the right track so to speak
minor:
check for copy-paste type of mistakes due to presence of greek letters, I noted one
namely in reference 106 "Diverting β- "
but there may be more, also some references are marked with an asterix,, why ?
Author Response
Referee 1:
Referee: “Excellent review that covers a wide array of literature concerning the interaction between various members of the FOX family and the WNT signalling pathway. In general, it is well balanced and as such will provide a great help for all that want to study this further.”
Response: Thank you!
Referee: “Being this the case I think (personally) that the reader would also be helped by being more critical concerning the literature. Now there is a "common themes and open questions" section which is laudable but could be strengtheedn for example by "cautionary notes" . For example, it is now suggested that because FOX family members bind to a similar DNA binding consensus many FOXO members could regulate expression of WNT modulators and therefore " the role of FOX proteins in the transcriptional regulation of Wnt pathway components may be underappreciated". However and alternatively one could argue that it is overestimated because many approach gene regulation by overexpression of FOX components and this may force binding to these same DNA elements.”
Response: The referee is correct in pointing out that most of the referenced studies – including work from our group – rely heavily on protein overexpression, which may confound results. I have now revised chapter 2.3. accordingly, to bring this and the following limitations to the readers’ attention and temper the conclusions drawn in this section.
Referee: “Also because e.g. FOXM1 regulates proliferation it should be considered that proliferation per se regulates many of the observed changes rather than these being FOXOM1 specific, same holds for FOXOs that can impose cell cycle arrest. Although have not gone to such detail through many of the referred papers I will not be surprised if several studies do not control for this possibility.”
Response: It is true that few studies referenced in this manuscript have considered that results could be an epiphenomenon of, for example, altered cell cycle regulation. I have added a cautionary note to this effect in the revised chapter 2.3.
Referee: “In addition, many studies lack details on the interaction e.g. between FOX and beta-catenin and for example the interaction between FOXM1 and beta-catenin has been suggested to at least involve the conserved FH domain. This may however also imply that if FOX proteins employ this mode of binding beta-catenin in general, this binding may inhibit FOX binding to DNA and hence transcriptional activity. So, could beta-catenin be considered a general FOXO repressor?”
Response: This is a great question, which I had not considered before! In fact, our group and others have observed that Ku70, which was recently shown to bind the FOXL2 forkhead box (PMID: 32332759), represses FOX-dependent transcription (PMIDs: 32858261, 20570964), so it is conceivable that beta-catenin has a similar effect. To my knowledge, this hypothesis has not been tested so far, so I would consider it premature to include it in this manuscript. However, our group is actively investigating the reciprocal regulation of Wnt signaling by FOX proteins, so we can hopefully answer this question conclusively in a future study. Regarding the interactions of beta-catenin with FOX proteins, I have added a schematic (new Figure 4) that summarizes what is currently known about interacting domains between these proteins.
Referee: “So if anything I would encourage the author to include a little more critical appraisal of literature to help new-comers to the field on the right track so to speak”
Response: Thank you again for the thoughtful and constructive comments!
Referee: check for copy-paste type of mistakes due to presence of greek letters, I noted one namely in reference 106 "Diverting β- "but there may be more, also some references are marked with an asterix,, why ?”
Response: Curiously, the asterisks are actually in the manuscript titles on the publisher’s website, and were imported automatically by the reference manager software. I have now corrected these and other mistakes in the reference list.
Reviewer 2 Report
This is a comprehensive review of the interaction between Fox transcription factors and Wnt pathways. The author has made significant contributions to this emerging intersect. This review will become a useful reference for readers interested in the Fox and Wnt pathways in biological processes especially cancer.
One of the strengths of this review is the in-depth description of how Fox proteins help stabilize beta-catenin and enhance interaction between beta-catenin and TCF/LEF. I feel that including illustrations of functional/structural domains of beta-catenin, Fox proteins (using FoxM1 as example) will make the discussion easier to follow. For Fox proteins that both stabilize beta-catenin and enhance catenin-TCF/LEF interaction, is the same domain involved in both processes? Why some Fox proteins function as inhibitors while others work as activators? Can the author offer some insights or hypotheses?
Author Response
Referee 2:
Referee: “This is a comprehensive review of the interaction between Fox transcription factors and Wnt pathways. The author has made significant contributions to this emerging intersect. This review will become a useful reference for readers interested in the Fox and Wnt pathways in biological processes especially cancer.“
Response: Thank you!
Referee: “One of the strengths of this review is the in-depth description of how Fox proteins help stabilize beta-catenin and enhance interaction between beta-catenin and TCF/LEF. I feel that including illustrations of functional/structural domains of beta-catenin, Fox proteins (using FoxM1 as example) will make the discussion easier to follow.”
Response: This is a good suggestion. I have now added an illustration of known interactions between beta-catenin and FOXM1, FOXO1, FOXG1, and TCF/LEF (new Figure 4), which will hopefully make this part of the discussion more readable.
Referee: “For Fox proteins that both stabilize beta-catenin and enhance catenin-TCF/LEF interaction, is the same domain involved in both processes?”
Response: To the best of my knowledge, this question has not been addressed in any of the studies.
Referee: “Why some Fox proteins function as inhibitors while others work as activators? Can the author offer some insights or hypotheses?“
Response: The most likely explanation at this point is that the unique regions of individual FOX proteins have evolved to recruit different interaction partners, notably including beta-catenin, that either promote or inhibit Wnt signaling. It is likely that this is used for fine-tuning Wnt signaling in tissues where sharply defined Wnt activity gradients are required, such as the developing brain or intestine. I have now expanded a bit more on this hypothesis in the revised chapter 2.3.